# Life Cycle Assessment of Carbon Capture by an Intelligent Vertical Plant Factory within an Industrial Park

**Haoyang Chen [1], Xue Dong [1,\*], Jie Lei [2], Ning Zhang [2], Qianrui Wang [2], Zhiang Shi [2] and Jinxing Yang [2]**

[1] China-UK Low Carbon College, Shanghai Jiao Tong University, Shanghai 200240, China; chenhaoyang1895@sjtu.edu.cn
[2] Volvo Cars Technology (Shanghai) Co., Ltd., Shanghai 201800, China
\* Correspondence: xue.dong@sjtu.edu.cn; Tel.: +86-136-3646-8642

**Abstract:** Bio-based carbon capture and utilization emerges as a critical pathway to mitigate carbon dioxide ($CO_2$) emissions from industrial activities. Within this context, plant factories become an innovative solution for biological carbon capture within industrial parks, fed with the substantial carbon emissions inherent in industrial exhaust gases to maximize their carbon sequestration capabilities. Among the various plant species suitable for such plant factories, *Pennisetum giganteum* becomes a candidate with the best potential, characterized by its high photosynthetic efficiency (rapid growth rate), perennial feature, and significant industrial value. This paper studies the feasibility of cultivating *Pennisetum giganteum* within an intelligent plant factory situated in an industrial park. An automated and intelligent plant factory was designed and established, in which multiple rounds of *Pennisetum giganteum* cultivations were performed, and life cycle assessment (LCA) was carried out to quantitatively evaluate its carbon capture capacity. The results show that the primary carbon emission in the plant factory arises from the lighting phase, constituting 67% of carbon emissions, followed by other processes (15%) and the infrastructure (10%). The absorption of $CO_2$ during *Pennisetum giganteum* growth in the plant factory effectively mitigates carbon emissions from industrial exhaust gases. The production of 1 kg of dry *Pennisetum giganteum* leads to a net reduction in emissions by 0.35 kg $CO_2$ equivalent. A plant factory with dimensions of 3 m × 6 m × 2.8 m can annually reduce carbon emissions by 174 kg, with the annual carbon sequestration per unit area increased by 56% compared to open-field cultivation. Furthermore, large-scale plant factories exhibit the potential to offset the carbon emissions of entire industrial parks. These findings confirm the viability of bio-based carbon capture using intelligent plant factories, highlighting its potential for carbon capture within industrial parks.

**Keywords:** carbon capture; plant factory; *Pennisetum giganteum*; life cycle assessment; environmental impact; industrial park

## 1. Introduction

To achieve the objectives outlined in the Paris Agreement [1], which aims to limit the average global temperature increase to well below 2.0 °C, it is imperative to make substantial reductions in worldwide greenhouse gas (GHG) emissions. The industrial sector, accountable for 33% of global anthropogenic GHG emissions [2], plays a fundamental role in achieving these overarching global climate and net-zero objectives. Carbon capture and utilization stands as a pivotal pathway for industrial decarbonization. Traditionally, carbon capture comprises three primary methods: 1. absorption, including physical absorption (using chemically inert solvents like diethyl ethers of polyethylene glycol, methanol, and N-methyl-2-pyrrolidone [3]), chemical absorption (with solvents such as amine-based alkali solvents, the most widely adopted method currently [4]), and biological absorption (forestation, microalgae, energy crops, etc. [5]); 2. adsorption (employing solid, rigid adsorbents [3]); 3. membrane separation (utilizing thin layers of organic or porous inorganic

materials [6]). A brief analysis of the advantages and disadvantages of various typical carbon capture technologies is presented in Table 1.

**Table 1.** Comparison of different carbon capture technologies.

| Technology | | Advantages | Disadvantages | Reference |
|---|---|---|---|---|
| Absorption | Physical absorption | High capacity at low temperature and high pressure<br>Cheaper solvent | Low capacity at high temperature and low pressure<br>High energy consumption | [3,5] |
| | Chemical absorption | High capacity at low $CO_2$ pressure<br>Thermally stable<br>Mature technology | High solvent loss due to evaporation<br>Absorbent degradation<br>High operating cost | [5,7] |
| Adsorption | Chemical adsorbents | Work at high temperature<br>High capacity | Negative effect of moisture<br>High energy consumption | [5,8] |
| | Physical adsorbents | High capacity at low temperature and high pressure<br>Low waste generation | Low $CO_2$ selectivity<br>Capacity decreases with temperature<br>Normally require high pressure | [5,9] |
| Membrane technology | | High separation efficiency<br>Low waste generation<br>Relatively low operation cost | High manufacturing cost<br>Relatively low separation selectivity<br>Negative effect of moisture | [6,10] |
| $CO_2$ storage | Geological sequestration | Huge storage capacity<br>Replenish depleted oil/gas reserves | High operational cost<br>Risk of $CO_2$ leakage and environmental contamination<br>Specific geomorphic structure requirement | [11,12] |
| | Oceanic injection | Huge $CO_2$ storage capacity | Cost intensive<br>Potential threat to marine life | [13,14] |
| Biological absorption | Forestation | Huge $CO_2$ storage capacity<br>No hazards of chemicals | Long time requirement<br>Large area requirement<br>May affect biological diversity | [15,16] |
| | Microalgae | Highly efficient<br>Faster growth rate than plants | Large facility requirement<br>Sensitive to other flue gas components and contamination | [7,17] |
| | Energy crop | Low cost input<br>Combined with industrial production<br>Co-production of feed, biofuel, and value-added products | Low carbon sequestration efficiency<br>Low land utilization rate<br>Immaturity of crop energy utilization | [18,19] |

Among these carbon capturing techniques, bio-based technology is one of the most environmentally friendly approaches, as it does not rely on industrial chemicals or materials as the capture medium. The biological absorption method involves cultivating terrestrial plants or autotrophic microorganisms in controlled environments and utilizing their photosynthesis to capture $CO_2$. Photosynthetic reactions are considered as a natural process with the capacity to generate valuable biomass as products. In a controlled environment, organisms proficient in photosynthetic reactions may offer an avenue to reduce emissions in an economically and environmentally sustainable manner [20]. Moreover, the integration of bioenergy with carbon capture has the potential to generate carbon-negative heat and power [21]. Currently, algae-based carbon capture research has drawn significant attention. For instance, Ramaraj et al. [22] simulated natural water conditions to cultivate algae and assess their carbon fixation potential. Judd et al. [23] designed photobioreactors to study the absorption of $CO_2$ and nutrient assimilation by algae. Wei et al. [24] comprehensively evaluated the microalgae harvesting strategies for biogas production via anaerobic digestion by comparative life cycle assessment. Nevertheless, the practical industrial application of microalgae encounters significant challenges. This is because microalgae cultivation demands either open raceway ponds or closed photobioreactors, the former of which is space-demanding, while the latter requires complicated facilities, and both of which are subject to strict environmental prerequisites [25]. Furthermore, prominent barriers also emerge during the industrialization and commercialization of microalgae-based technolo-

gies due to high costs and energy consumption [17]. For example, dehydration, as the initial step in algae product utilization, is substantially energy-intensive due to the high water content of microalgae. In contrast, the artificial cultivation and resource utilization of terrestrial plants exhibit a higher level of maturity and offer substantial potential for carbon capture in industrial environments.

Plant factories emerge as a potential platform for terrestrial plant-based carbon capture within industrial parks. A plant factory with artificial lighting (PFAL) is a closed plant production system that utilizes artificial lighting to cultivate crops and provides precise control over environmental parameters such as temperature, illumination, humidity, $CO_2$ concentration, etc. By controlling factors such as light quality and nutrient supply, PFALs can optimize plant growth, development, and nutritional value. The multi-tiered cultivation system facilitates space utilization while allowing for customization of plant morphology and metabolite composition to meet specific requirements [26]. Therefore, the growth of plants in a PFAL is self-sufficient and unaffected by the external environment, making it a promising solution to overcome the limitations of traditional agriculture, achieving year-round crop production and a high space utilization rate. Meanwhile, high carbon emission industrial factories possess the potential to provide a favorable environment for plant growth. The growth of plants can be used to absorb and fix $CO_2$ through photosynthesis, lowering the concentration of $CO_2$ in exhaust gases and achieving carbon capture [27], supplying biomass and bioenergy to the entire biosphere [28]. Some scholars also discovered that utilizing industrial production waste as raw materials for plant factories could lead to improved environmental benefits [29,30]. Furthermore, mature plants can be processed into bio-based fuels or materials that can be effectively used or recycled back into the industrial chain.

Numerous studies have demonstrated that implementing suitable internal environmental control strategies in plant factories can significantly enhance plant growth. For instance, Liang et al. [31] have demonstrated that peppers can grow better under appropriate artificial strategies including light and temperature control in PFALs. Zhang et al. [32] have built a closed PFAL and found that the production of the plants increased by 20–25%, and the plants fixed a considerable amount of $CO_2$ by increasing the concentration of $CO_2$ in the environment to 1000 ppm. Chowdhury et al. [33] established five environmental conditions in a plant factory for comparison and discovered the best $CO_2$ concentration, humidity, and temperature ranges for kale growth and total glucosinolate content, respectively. Chen et al. [34] have found that elevated $CO_2$ concentrations in a PFAL positively influenced lettuce growth, light-use efficiency, and yield, demonstrating the potential of $CO_2$ enrichment in PFAL systems. Zhiwei T et al. [35] investigated the effects of LED lighting arrangement, light source type, and switching intervals on plant growth in artificial light plant factories, aiming to achieve an optimal cultivation strategy. Y. Kikuchi et al. [36] compared different plant factory models, including those with sunlight and artificial light, as well as traditional agricultural methods. They analyzed the consumption of nitrogen, phosphorus, potassium, water, and greenhouse gas emissions and identified the advantages of plant factories in crop production and energy efficiency. Research findings also demonstrate that the utilization of intelligent technology in plant factories improves sustainability performance by augmenting production productivity, product quality, annual crop yield, and resource use efficiency [37,38]. In conclusion, plant factories can, to a certain extent, address the issues of low carbon sequestration efficiency and suboptimal land-use efficiency in traditional energy crop cultivation.

*Pennisetum giganteum* stands out as the optimal choice for carbon sequestration using plant factories within industrial parks. *Pennisetum giganteum*, a fast-growing perennial C4 grass, belonging to the genus Pennisetum and the Poaceae family, is native to eastern and northeastern African tropical regions. It has gained widespread popularity and has been cultivated in more than 80 countries worldwide and in over 30 provinces in China [39]. The adaptability and versatility of *Pennisetum giganteum* have positioned it as a promising candidate for sustainable biomass production and utilization in various applications, in-

cluding bioenergy production and environmental remediation [39]. *Pennisetum giganteum* has gained recognition as a potential source of alternative energy due to its high cellulose content, because this lignocellulosic energy crop can work as a second-generation biomass feedstock for bioethanol production. In the industrial context, *Pennisetum giganteum* demonstrates diverse applications including direct combustion, methane conversion, hydrogen production, fuel ethanol, biodiesel, briquetting fuel, biomass power generation, biomass gasification, industrial fiber raw material, etc. [40–42]. One effective method for harnessing energy from *Pennisetum giganteum* involves mesophilic anaerobic digestion (MAD). During MAD, the materials of *Pennisetum giganteum* undergo degradation by the fermentation microflora within the methanogenic system, resulting in the production of volatile fatty acids (VFAs), methane, hydrogen sulfide, and other compounds [41,42]. Additionally, ongoing research explores the utilization of *Pennisetum giganteum* for hydrogen production through photofermentation. Under conditions involving the addition of photosynthetic bacteria, *Pennisetum giganteum* can generate hydrogen through enzymatic hydrolysis [43]. In general, *Pennisetum giganteum*, as a promising energy production resource, is a potentially viable option for carbon sequestration in plant factories.

Many studies have employed life cycle assessment (LCA) to evaluate the environmental impacts of various plants throughout their life cycles, including maize, willow, alfalfa, straw, grass-clover, ryegrass, and winter wheat, among others [44–46]. LCA is also employed for comparing various agricultural production methods. With the emergence of vertical farming and plant factories, LCA has gained widespread attention for assessing the carbon footprint of these systems. The framework of vertical farming, along with its characteristics, advantages, and disadvantages, has been elucidated by researchers. Martin et al. [47] discovered by LCA that electricity demand for lighting was a major source of environmental impact in vertical hydroponic systems. Hallikainen et al. [48] found that cultivating under artificial lighting also increased energy consumption while improving land utilization and concluded that its environmental advantages would gradually become apparent as energy sources become cleaner in the future. Prior research in this field has primarily focused on plant factories for traditional vegetables instead of industrial biofuels or biomaterials. This study seeks to fill this gap by performing a comprehensive study of the *Pennisetum giganteum* plant factory and conducting an environmental impact analysis using LCA principles.

## 2. Materials and Methods

### 2.1. Design of an Intelligent Plant Factory

The conceptual framework for a plant factory integrated within an industrial park is illustrated in Figure 1. In this process, exhaust gases from industrial facilities are introduced into the plant factory as a carbon source, and the plant factory is powered by renewable electrical power (solar PV) and is fed with reclaimed water from the industrial park. Subsequently, the plant factory cultivates high-value industrial crops that actively sequester $CO_2$. As these crops are harvested, they are processed into biofuels like bioethanol and biodiesel or transformed into bio-based materials such as bioplastics and rubber. These products can be reintegrated into industrial production processes. It is important to note that the efficiency of carbon capture within the industrial chain is intricately linked to the performance of the plant factory system.

To validate this concept, this research has successfully designed and constructed an intelligent plant factory, as is shown in Figure 2, and the hardware architecture of the plant factory is shown in Figure 3. The sensor system serves as a critical data collection platform for essential environmental parameters, including light intensity, $CO_2$ concentration, soil pH, air temperature and humidity, soil temperature and humidity, image information, etc. Table 2 presents the specifications for various sensors and measurement instruments in the plant factory. The central control unit governs the operation of various devices including the fan, the irrigation system, lighting system, the air conditioner, and the sterilization equipment based on feedback from the sensor system and programmed strategies.

For instance, it regulates the introduction of industrial exhaust gases depending on factory conditions (with fans activated during factory operating hours). It also controls the timing of lighting and sterilization (12 h per day with a predetermined time interval). If the environmental temperature exceeds predetermined limits, the controller activates the air conditioning system (20–35 °C during the summer and 10–20 °C during the winter). Furthermore, the controller ensures soil moisture within predefined ranges by regulating irrigation frequency (maintaining soil moisture between 60 and 80%).

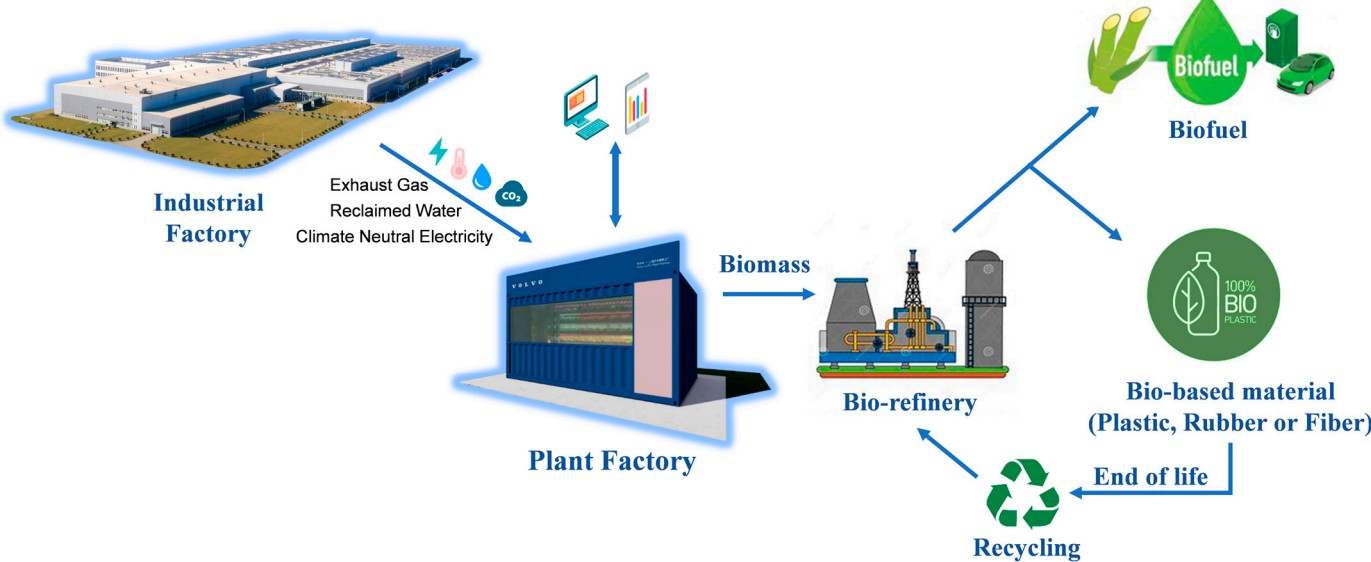

**Figure 1.** Intelligent vertical plant factory in the industrial chain.

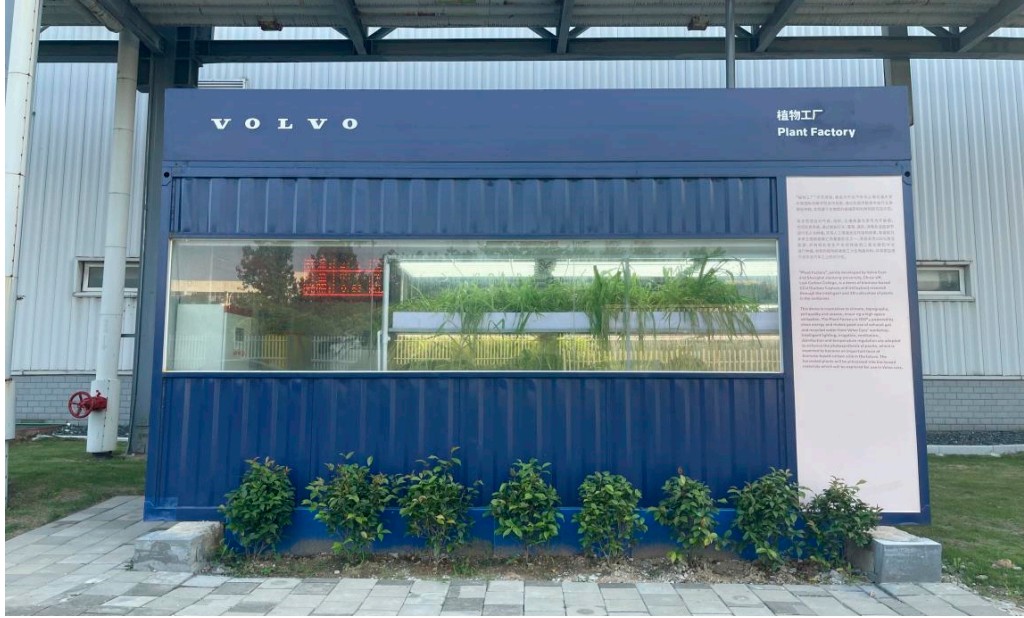

**Figure 2.** Exterior of the intelligent plant factory.

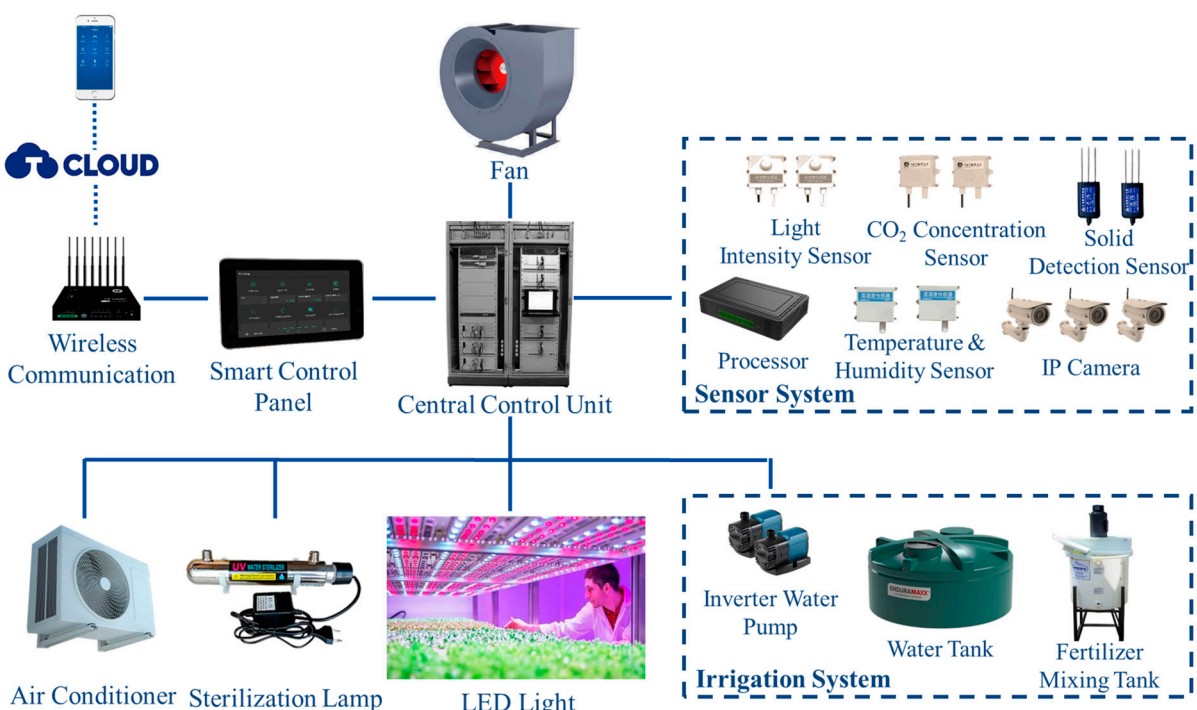

**Figure 3.** System architecture of the intelligent plant factory.

**Table 2.** Parameters of sensors and measurement instruments.

| Instrument | Measurement Range | Resolution | Accuracy |
|---|---|---|---|
| Air Temperature Sensor | −20~50 °C | 0.1 °C | ±0.5 °C |
| Air Humidity Sensor | 0~100% RH | 0.1% RH | ±0.5% (at 60% RH, 25 °C) |
| $CO_2$ Concentration Sensor | 0–5000 ppm | 1 ppm | ±(50 ppm + 3% F·S) |
| Light Intensity Sensor | 0.1–10 klux | 0.01 klux | ±4% (at 25 °C) |
| Soil Temperature Sensor | −40~80 °C | 0.1 °C | ±0.5 °C |
| Soil Humidity Sensor | 0~100% RH | 0.1% RH | ±3% (at 60% RH, 25 °C) |
| Soil pH Sensor | 3~9 | 0.01 | ±0.5 |
| Electrical Conductivity Sensor | 0–2000 μs/cm | 1 μs/cm | ±5% |
| Liquid Level Sensor | — | 1 cm | — |
| Water Meter | — | 1 L | — |
| Electricity Meter | — | 0.1 kWh | — |
| Electronic Scale | 0–50 kg | 1 g | ±10 g |

The control interface of the plant factory is shown in Figure 4. It enables operators to manage various devices, including the irrigation system, lighting system, sterilization lamp, and air conditioning. Predetermined operation strategies can be implemented to generate an optimal growth environment for plants, allowing for preset and closed-loop control of parameters such as timing for lighting, irrigation frequency and volume, fertilization frequency and amount, temperature and humidity ranges, etc. Moreover, operational and sensor data from each device are systematically recorded, enabling remote data loading and analysis. The plant factory incorporates advanced sensors, intelligent controls, remote connectivity, and a user-friendly interface for on-site and remote control, allowing online monitoring and real-time, precise, efficient, and unmanned operations.

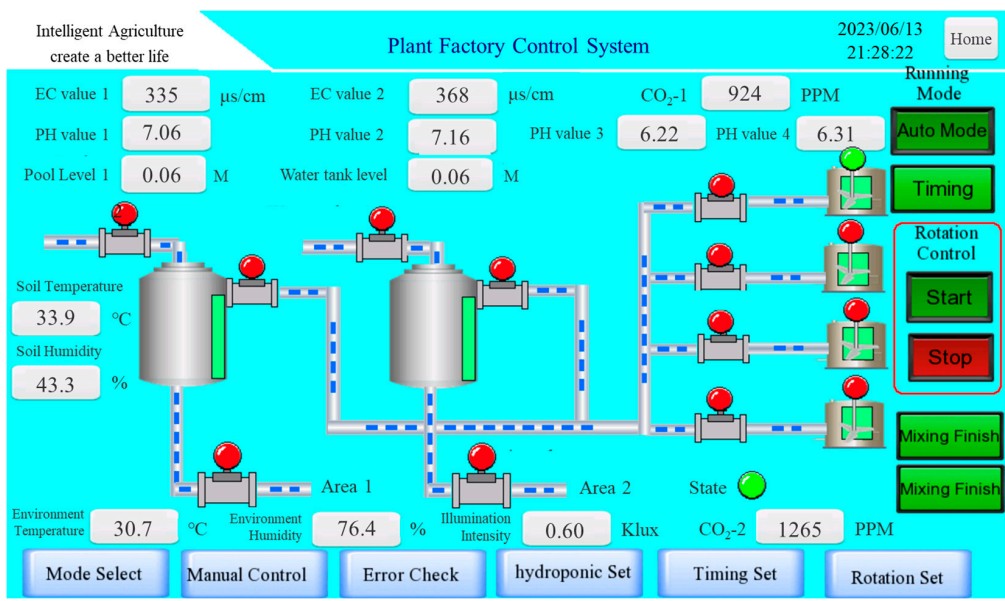

**Figure 4.** Control interface of the intelligent plant factory.

### 2.2. Establishment of Plant Factory with Exhaust Gases Introduction

Among the different stages of automobile production, the painting process is recognized as one of the major contributors to carbon emissions. By employing advanced treatment techniques, the concentration of harmful substances in the exhaust gases from the painting workshop is effectively reduced to comply with stringent atmospheric emission standards. In a painting workshop, volatile organic compounds (VOCs) constitute the primary pollutants, and they are effectively managed through a combined treatment approach involving zeolite rotating wheel adsorption and RTO (Regenerative Thermal Oxidizer). Natural gas serves as the combustion assistant during the incineration process, converting VOCs into non-polluting constituents such as $CO_2$ and $H_2O$. This treatment has removed over 90% of pollutants and meets emission standards, as indicated in Table 3. However, it is important to note that the incineration process results in increased $CO_2$ emissions, resulting in a $CO_2$ concentration 4–5 times higher than atmospheric levels. Consequently, these emissions serve as a suitable industrial carbon source for the plant factory.

**Table 3.** Pollutant concentration after treatment in the exhaust gas of the painting workshop (the average results from 3 independent samples).

| Items | Emission Concentration (mg/m$^3$) | Emission Limit (mg/m$^3$) |
|---|---|---|
| Particulate Matter | 1.5 | $\leq$30 |
| 1,3,5-Trimethylbenzene | <0.01 | - |
| 1,2,4-Trimethylbenzene | <0.01 | - |
| Non-Methane Total Hydrocarbons (as Carbon) | 2.9 | $\leq$60 |
| Toluene | <0.004 | - |
| Xylene | 0.014 | - |
| Ethyl Acetate | <0.005 | - |
| $CO_2$ | 2034 ppm | |

In this study, a specialized pipeline was developed to facilitate carbon capture within the plant factory. Exhaust gases originating from the painting workshop were introduced into the plant factory through an external gas pipeline connected to the workshop's chimney, as illustrated in Figure 5a. The pipeline, with a diameter of 110 mm and a length of 70 m, ducted the $CO_2$-enriched exhaust gas into the plant factory. The gas intake was driven by a

5.5 kW high-pressure fan, and a manual valve allowed precise control over the gas flow rate into the container.

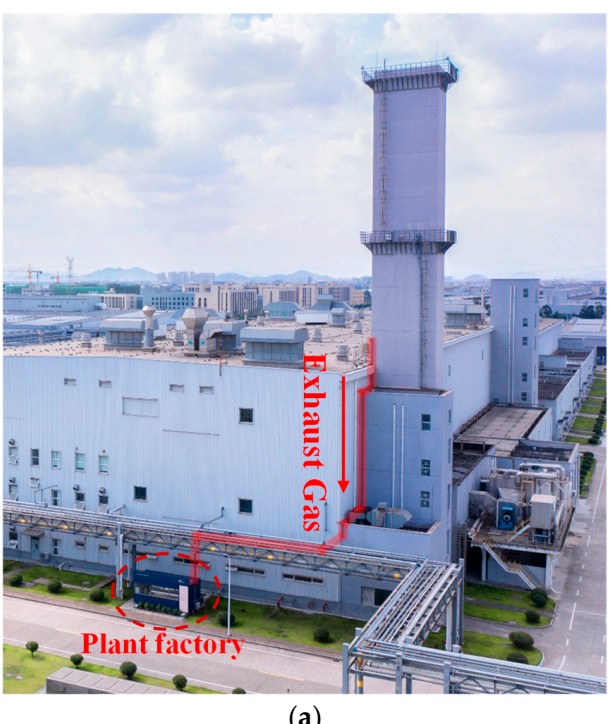 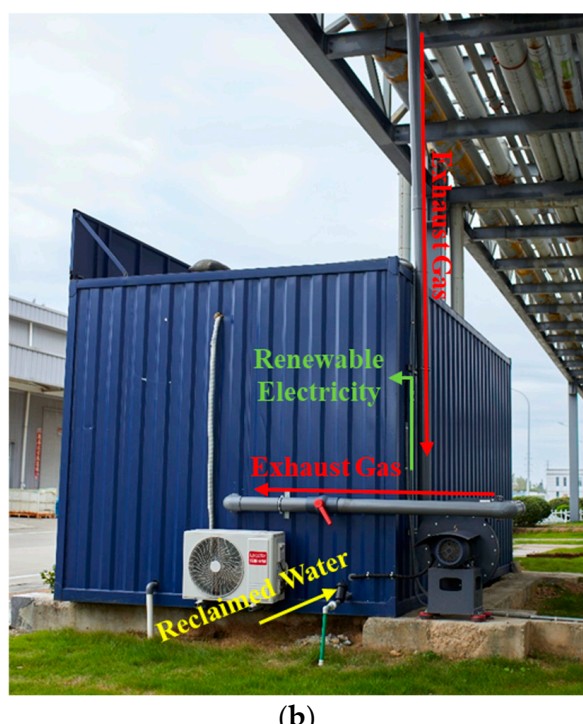

(**a**)          (**b**)

**Figure 5.** Industrial exhaust gases are directed into the plant factory via the chimney with the gas pipelines highlighted in red. The green and yellow lines represent electrical and water pathways, respectively. (**a**) Front view and (**b**) side view of the plant factory and exhaust gas pipeline.

In addition, the plant factory is powered by a solar photovoltaic system, and water is sourced from the reclaimed water treatment facility in the industrial park, as illustrated in Figure 5b. It is noteworthy that the industrial park primarily sources its electricity from solar photovoltaic panels within the park, with the remainder purchased as solar power, ensuring that all electricity used is derived from photovoltaic generation.

### 2.3. Selection of Plants

In the context of industrial applications, especially for the automotive supply chain, specific plant species have demonstrated their potential for the production of bio-fuel or bio-based materials [49]. This study comprehensively evaluated various plant species based on five key criteria, including carbon sequestration capacity, temperature requirements, space requirements, growth rate, and application value. The assessment encompassed plants currently in use or with prospective applications within Volvo's automotive supply chain. The summarized evaluation results in Table 4 underscore *Pennisetum giganteum* as an exceptional choice due to its remarkable carbon sequestration capacity, rapid growth rate, and manageable cultivation requirements. Particularly noteworthy is the high carbon sequestration efficiency of *Pennisetum giganteum*, attributed to its C4 photosynthetic pathway, which exhibits an absorption efficiency for $CO_2$ that surpasses ordinary C3 crops by a factor of 4–7.46 [50]. Consequently, this study selects *Pennisetum giganteum* as the preferred plant for the plant factory.

**Table 4.** Assessment of suitability for different plants. (The selected plants are already applied or possessing potential application value within the supply chain of Volvo.)

| Plant | Carbon Sequestration Capacity | Temp Range (°C) | Min Harvest Height (m) | Growth Cycle (month) | Harvest Frequency (time/year) | Regeneration Capacity | Applied Organ | Product | Industrial Application |
|---|---|---|---|---|---|---|---|---|---|
| *Pennisetum giganteum* [51,52] | C4 plant | 10–35 | 0.5 | 1–3 | 3–4 | Perennial | Stem leaf | Cellulosic ethanol/polymer composite | Processed for cellulose ethanol or used in the manufacturing of composite materials. |
| Bamboo [53] | C3 plant with high carbon sequestration efficiency | 8–36 | 2 | >12 | 1 | Perennial | Stem | Bamboo fiber composite | Extracted from culms using mechanical or chemical methods for composite production. |
| Russian dandelion [54] | C3 plant | 15–35 | 0.2 | 6 | 1 | Perennial | Root | Rubber | Produce a milky fluid in roots, containing a high-quality rubber. |
| Castor oil plant [55] | C3 plant | 14–36 | 1.5 | 6 | 1 | Perennial | Seed | Polymer composite | Castor oil can be a source for polymers such as polyurethanes, polyesters, polyamides, and epoxy-polymers. |
| Fungi [56,57] | - | 5–35 | 0.2 | 1 | 8–12 | Annual | Mycelium | Polymer composite or mycelium-based leather | Transform into a resilient natural composite with controlled properties through chemical and heat treatments. |
| Hemp [58] | C3 plant | 19–25 | 1 | 3–4 | 1 | Annual | Stem | Hemp fiber composite | Wide-ranging applications in the automotive, electrical, construction, and packaging sectors. |
| Pineapple Tree [59] | C3 plant | 20–30 | 0.6 | 18–24 | 1 | Perennial | Leaf | Polymer composite | Pineapple leaf fiber can be applied in the making of reinforced polymer composites. |
| Giant Reed [60] | C3 plant | 10–35 | 1 | >12 | 1 | Perennial | Stem leaf | Polymer composite | Obtained through mechanical processes and used for composite materials production. |
| Cactus [61] | C3 plant | 20–30 | 1 | >12 | 1 | Perennial | Leaf | Cactus-based leather | Employed in the manufacturing of various products including car interiors. |

*2.4. Cultivation of Pennisetum giganteum in and out of Plant Factory*

2.4.1. Planting Experiment and Measurement

The growth experiment of *Pennisetum giganteum* was conducted from October 2022 to October 2023. During the growth phase of *Pennisetum giganteum*, measurements were taken every 2 days for the experimental group. Employing a systematic sampling method, 10 plants of *Pennisetum giganteum* were selected at equal intervals. Heights of the plant were recorded, excluding plants with obvious growth abnormalities, and the averages were calculated and documented. At the harvesting stage, the unit area yield was calculated based on the fresh weight of harvested *Pennisetum giganteum* and the harvested area for the experimental group.

2.4.2. Statistical Analysis of Planting Experiments

The results were examined by ANOVA with SPSS 26 statistical software. One-way ANOVA and Duncan test were used, and a threshold of $p < 0.05$ was considered to indicate a significant difference. For each round of planting experiment, the heights of each group's 10 samples are used for analysis. For multi-round planting experiments, the average height and average yield of each round are used for analysis.

2.4.3. Experiments and Results for Outdoor Plantation

A comparative outdoor cultivation experiment was conducted on *Pennisetum giganteum* to investigate the effects of five different cultivation strategies, including soil substrate depth, light intensity, and nutrient requirements. The purpose was to examine the growth characteristics of the plants under natural outdoor light conditions and then compare them with those cultivated indoors in the controlled environment of an artificial light-based plant factory.

The growth condition of the outdoor experiments is illustrated in Figure 6, and the statistical results of the outdoor experiments are presented in Table 5. In similar outdoor environmental conditions, no significant difference in growth performance was observed between Group O-1 and Group O-4, indicating that *Pennisetum giganteum* can thrive within contained planter conditions. Furthermore, the comparison of Group O-4 and Group O-5 showed that *Pennisetum giganteum* demonstrated superior growth under high sunlight intensity conditions in comparison to low sunlight intensity conditions. Furthermore, the experimental groups that received nutrient supplementation exhibited enhanced growth in contrast to the groups without nutrient supplementation, and both urea and compound fertilizer can promote the growth of *Pennisetum giganteum*. These findings emphasize the importance of appropriate lighting and proper nutrient supplementation for the proper cultivation of *Pennisetum giganteum*.

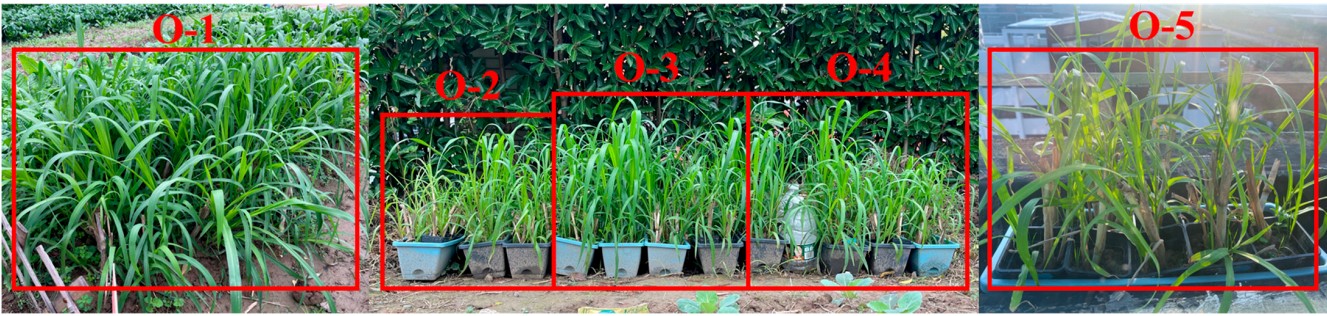

**Figure 6.** Growth status of *Pennisetum giganteum* from outdoor cultivation experiment.

**Table 5.** Cultivation strategies and results for outdoor experiment of *Pennisetum giganteum*.

| Group | Planting Environment | Planting Duration | Light Condition | Nutrient Solution | Height (cm) | Yield (kg/m$^2$) |
|---|---|---|---|---|---|---|
| O-1 | Open Field | 3 months | Sunlight | Compound Fertilizer | 76 ± 7.1 [a] | 8.3 |
| O-2 | Planters | 3 months | Sunlight | None | 65 ± 3.1 [b] | 7.2 |
| O-3 | Planters | 3 months | Sunlight | Urea Fertilizer | 74 ± 4.8 [a] | 8.3 |
| O-4 | Planters | 3 months | Sunlight | Compound Fertilizer | 72 ± 4.9 [a] | 8.2 |
| O-5 | Planters | 3 months | Semi-shading sunlight | Compound Fertilizer | 42 ± 5.9 [c] | 4.7 |

Note: Results of heights are shown as mean ± standard deviation. Different superscripts denote significant differences at a 95% confidence level ($p < 0.05$), and the order of a, b and c represents the mean values from largest to smallest. Heights with the same letter do not differ significantly according to the Duncan test.

### 2.4.4. Experiments and Results for Plant Factory

*Pennisetum giganteum* exhibits the ability to promptly initiate successive growth cycles following each harvest within an optimal environmental setting. Although it has the potential to achieve an impressive height of up to 7.08 m [51], attaining maximum height is unnecessary in the context of a plant factory. Therefore, a thorough analysis of the optimal frequency for harvesting is necessary to maximize the productivity of *Pennisetum giganteum* within an annual production cycle in the plant factory. Four distinct planting strategies were applied, which involved varying irrigation frequencies and nutrient application rates, aiming to investigate the effects of these factors on the growth rate and quality of *Pennisetum giganteum*. Table 6 presents the cultivation strategies for each group. Figure 7 shows the growth status of *Pennisetum giganteum* in the plant factory, and Figure 8 illustrates the change in the height of the plants over time. It can be seen that *Pennisetum giganteum* exhibits a period of accelerated growth between days 30 and 40, during which its height ranges approximately from 40 to 59 cm. Subsequently, the growth rate gradually decelerates. *Pennisetum giganteum* of this height range can meet the height restrictions of multi-tier cultivation racks. Moreover, after harvesting, it can rapidly initiate a new growth cycle, allowing for continuous cultivation. Therefore, it is recommended to implement a harvesting cycle every 35–40 days to optimize production efficiency and achieve a maximum of 10 harvests per year within an appropriate temperature range. It can be observed from Table 6 that the plant height of Group D exhibits significant differences compared to the other three groups. These findings suggest that the irrigation frequency of 4 times per day and the nutrient application frequency of once every 5 days will enhance the growth of *Pennisetum giganteum*. Throughout the three-month cultivation period, the *Pennisetum giganteum* exhibits an average height increase of 50% compared to outdoor cultivation, accompanied by a 33% enhancement in yield.

**Table 6.** Cultivation strategies and results from 3-month plant factory experiment.

| Group | Planting Duration | Irrigation Frequency | Fertilizing Rules | Height (cm) | Yield (kg/m$^2$) |
|---|---|---|---|---|---|
| A | 3 months | 2 min/d | After growing 20 days, every 8 d | 88 ± 7.4 [c] | 9.6 |
| B | 3 months | 2 min/d | After growing 15 days, every 8 d | 94 ± 3.9 [b] | 9.9 |
| C | 3 months | 4 min/d | After growing 15 days, every 8 d | 95 ± 5.4 [b] | 10.1 |
| D | 3 months | 4 min/d | After growing 15 days, every 5 d | 114 ± 6.4 [a] | 11.6 |

Note: Results of heights are shown as mean ± standard deviation. Different superscripts denote significant differences at a 95% confidence level ($p < 0.05$), and the order of a, b and c represents the mean values from largest to smallest. Heights with the same letter do not differ significantly according to the Duncan test.

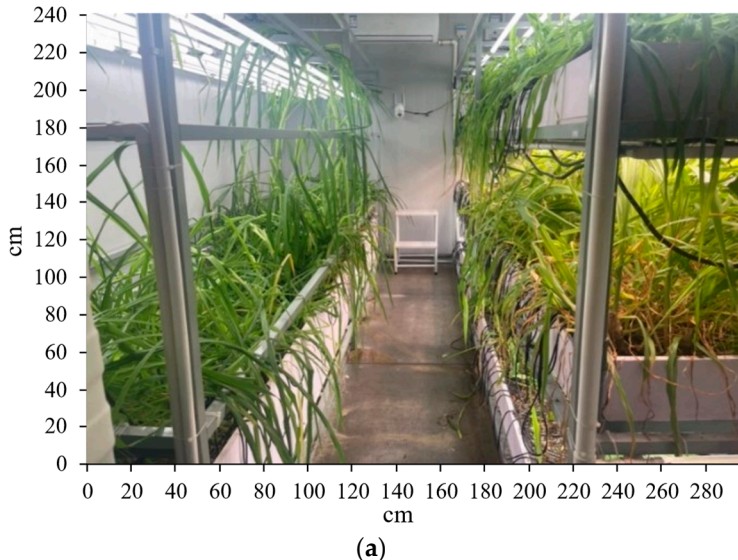
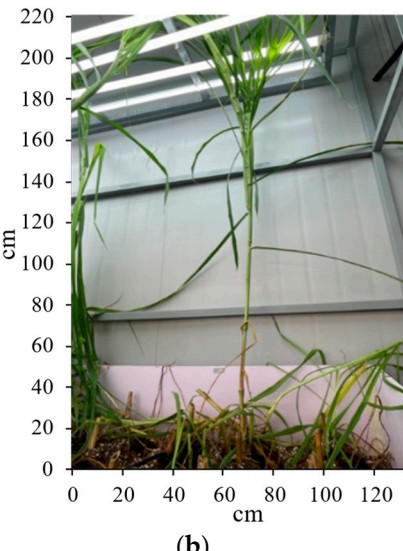

(**a**)　　　　　　　　　　　　　　　　　　(**b**)

**Figure 7.** Growth status of *Pennisetum giganteum* in plant factory. (**a**) Inner view; (**b**) individual *Pennisetum giganteum* can reach a maximum height of up to 2.2 m within 3 months.

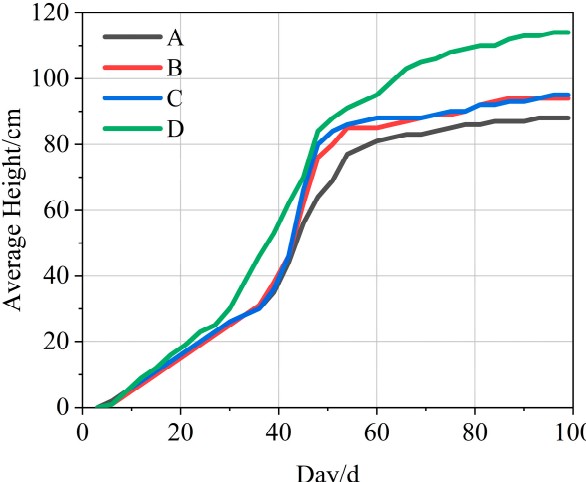

**Figure 8.** Height of *Pennisetum giganteum* over time in a plant factory.

The strategies from Group D were applied to further investigate the effects of light spectra and exhaust gases on the *Pennisetum giganteum* plant factory. The supplemental lighting period in the plant factory is from 7:00 to 19:00, covering 12 h each day. The experiments were performed under different light spectra; compared to plants grown under full-spectrum lighting, those exposed to the red–blue light spectrum demonstrated a 4% increase in average height and a 5% increase in yield, which suggest that the *Pennisetum giganteum* responds favorably to the specific red–blue light spectrum employed in the plant factory. Industrial exhaust gases are introduced into the plant factory to elevate $CO_2$ concentrations during the lighting period, as depicted in Figure 9. In the plant factory without exhaust gas introduction, the concentration of $CO_2$ decreases during photosynthesis, while it increases at night due to respiration processes. In plant factories with exhaust gas introduction, the lighting period is synchronized with the operational hours of the painting workshop. The introduction of exhaust gases helps to maintain $CO_2$ levels in the plant factory in the range of 1600–2000 ppm. The $CO_2$ concentration gradually decreases after ceasing the exhaust gas introduction at night. In full-spectrum and red–blue spectrum lighting conditions, the introduction of exhaust gases resulted in an average yield increase of 17% and 15%, respectively,

demonstrating a favorable growth-promoting effect of industrially generated $CO_2$-enriched exhaust gases.

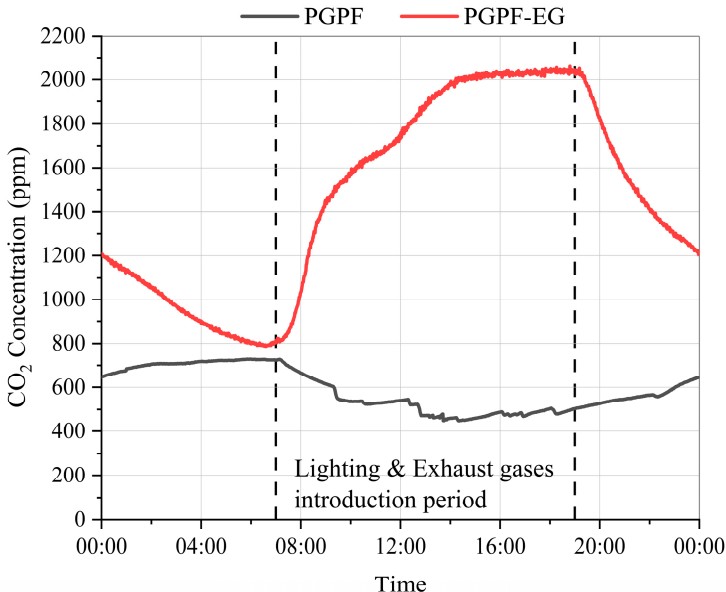

**Figure 9.** The $CO_2$ concentration over time in PGPF (*Pennisetum giganteum* plant factories without exhaust gases) (20-day averages) and in PGPF-EG (*Pennisetum giganteum* plant factories with exhaust gases introduced) (20-day averages).

Table 7 and Figure 10 present the strategies and results from five consecutive cycles of planting experiments. It can be seen that Group D-IV exhibits a significant difference compared to the other groups, achieving the highest yield, employing cultivation strategies involving 12 h of red–blue light exposure, 4 min of daily drip irrigation, and fertilization every 5 days. Additionally, maintaining $CO_2$ concentrations at 1600–2000 ppm during the lighting period through the introduction of industrial exhaust gases enhances the yield to 6 kg/m$^2$. For a plant factory of 6 m × 3 m × 2.8 m in size (effective cultivation area of 36 m$^2$), an annual production of 2160 kg can be achieved for 10 growth cycles.

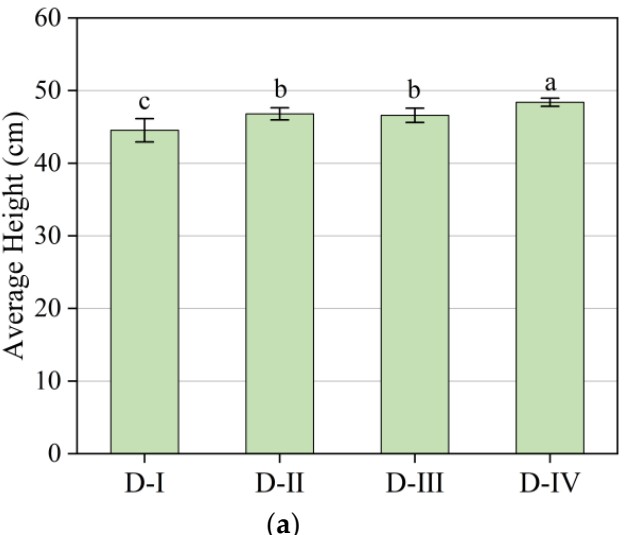

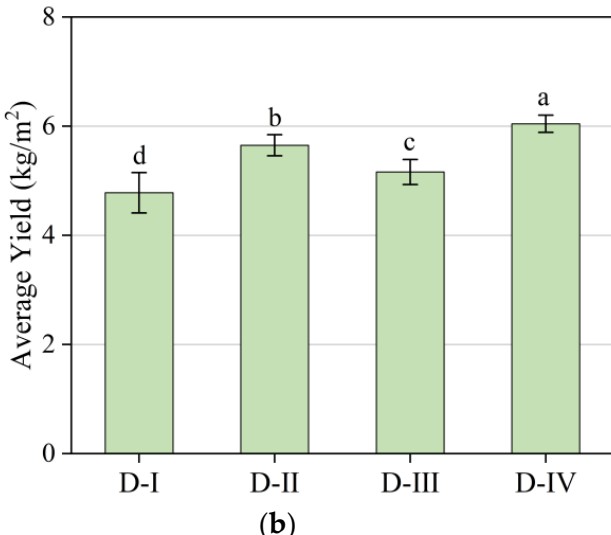

**Figure 10.** *Pennisetum giganteum* plant factory experiment results from five rounds of cultivation: (**a**) average height; (**b**) average yield. Different letters indicate differences at the confidence level of 95% ($p < 0.05$).

**Table 7.** Cultivation strategies and results from five rounds of planting of plant factory experiments.

| Group | Planting Duration | Lighting Conditions | Exhaust Gases Introduction | Average Height (cm) | Average Yield (kg/m²) |
|---|---|---|---|---|---|
| D-I | 35 days | Full spectrum | no | 45 ± 1.6 [c] | 4.8 ± 0.4 [d] |
| D-II | 35 days | Full spectrum | yes | 47 ± 0.8 [b] | 5.7 ± 0.2 [b] |
| D-III | 35 days | Red–blue light | no | 47 ± 1.0 [b] | 5.2 ± 0.2 [c] |
| D-IV | 35 days | Red–blue light | yes | 48 ± 0.6 [a] | 6.0 ± 0.2 [a] |

Note: Results of heights and yields are shown as mean ± standard deviation. Different superscripts in the same column denote significant differences at a 95% confidence level ($p < 0.05$), and the order of a, b, c and d represents the mean values from largest to smallest. Heights with the same letter do not differ significantly according to the Duncan test.

## 2.5. Carbon Capture Capability of the Pennisetum giganteum Plant Factory

In order to quantify the carbon capture potential of the plant factory, moisture quantification and elemental analysis was performed on the harvested *Pennisetum giganteum*. The harvested biomass was processed through a series of procedures, including cutting, thorough cleaning, and representative sampling. The collected samples were then carefully dried in a controlled environment at a constant temperature of 70 °C for a period of 72 h until achieving complete desiccation, thereby enabling the determination of the moisture content of the *Pennisetum giganteum* biomass. The analysis revealed an average moisture content of 77% for the harvested plant. Subsequently, the dried samples were finely ground and sieved to ensure homogeneity and uniformity of particle size. Elemental analysis, employing an elemental analyzer (Elementar vario EL CUBE, manufactured by ELEMENT, Berlin, Germany, test conditions with a combustion tube temperature of 1150 °C and a reduction tube temperature of 850 °C), was conducted to quantitatively determine the elemental composition in mass fraction. Three rounds of *Pennisetum giganteum* sampling were conducted, with each round subjected to two experimental measurements. The average results from these experiments are presented in Table 8.

**Table 8.** Results of elemental mass fraction analysis for dry *Pennisetum giganteum*.

| Sample | N (%) | C (%) | H (%) | S (%) |
|---|---|---|---|---|
| 1 | 3.78 | 40.93 | 5.28 | 0.25 |
| 2 | 3.86 | 40.03 | 5.35 | 0.23 |
| 3 | 3.81 | 41.98 | 5.12 | 0.39 |
| Average | 3.82 | 40.98 | 5.25 | 0.29 |

To quantify the carbon fixation capability during the growth process of *Pennisetum giganteum*, it is assumed that carbon capture from soil and nutrients is minor and can be neglected, as no carbon contents have been added to the nutrients and soil during or before the plantation. It was also worth noting that other carbonaceous gases, such as VOCs, are negligible in the exhaust gas directed to the plant factory, as VOCs are combusted before being exhausted to the chimney. It was therefore considered that the primary source of carbon in the harvested *Pennisetum giganteum* is $CO_2$ from the exhaust pipe. The quantity of $CO_2$ sequestered by the plant during its growth cycle is calculated as Equation (1):

$$m_{CO_2} = m_{fresh} \cdot (1 - \theta_{water}) \cdot \eta_C \cdot M_{CO_2} / M_C \tag{1}$$

where $m_{CO_2}$ denotes the mass of $CO_2$ assimilated and fixed by the plant. $m_{fresh}$ denotes the fresh weight of the plant. $\theta_{water}$ denotes the moisture content of the plant. $\eta_C$ corresponds to the carbon content as a percentage in the plant's dry matter. $M_{CO_2}$ denotes the molar mass of $CO_2$. $M_C$ denotes the molar mass of carbon.

### 3. Life Cycle Assessment of the *Pennisetum giganteum* Plant Factory

*3.1. Life Cycle Assessment Method*

This evaluation is performed following the ISO-standardized life cycle assessment (LCA) procedure. The ISO 14040 standard [62] defines LCA as a systematic and quantitative evaluation of the environmental impacts of a product system over its life cycle, including inputs and outputs of matter, energy, and pollutants [63], considering all stages from raw material extraction, manufacturing, distribution, use, and disposal. By assessing the environmental impact of a product throughout its life cycle, LCA provides a comprehensive and holistic approach to evaluate the potential environmental impacts and identify improvement opportunities for production and business models. LCA stands as a preeminent methodology extensively employed within the European Union across a spectrum of industries, encompassing agriculture, manufacturing, the energy sector, and various others [64]. Among different options of LCA software (https://pre-sustainability.com/solutions/tools/simapro, accessed on 10 November 2023), Simapro is a predominant choice, offering extensive databases like Ecoinvent, housing vast datasets across diverse processes. Consequently, this study employed Simapro 9.4 for LCA to evaluate the environmental impact of various cultivation systems and management practices.

*3.2. Goal and Scope Definition*

3.2.1. Goal

The goal of this study is to rigorously evaluate the resource utilization, $CO_2$ sequestration, and ensuing environmental benefits pertaining to the growth of *Pennisetum giganteum* under distinct conditions. This comprehensive assessment aims to study the potential for carbon sequestration within *Pennisetum giganteum* plant factories situated within an industrial park. To achieve this objective, a cradle-to-gate assessment was performed, including all upstream processes, with the boundary being the biomass produced by the plant factory.

3.2.2. System Boundary

The LCA boundary is described as shown in Figure 11. The infrastructure inputs for the plant factory involve diverse elements, including steel structures, PVC structures, glass, lighting equipment, sensors, and various other devices. The environmental impacts of these infrastructures will be evenly distributed over the operational lifespan of the plant factory. In terms of material inputs, vital considerations include plant seeds, plant growth medium, $CO_2$, water, and fertilizer. These materials form the core inputs driving the plant growth and development processes. Simultaneously, energy-intensive activities are accounted for, such as the operation of lighting systems, water pumps, air conditioning units, and ventilation systems. The life cycle assessment analysis extends beyond the cultivation phase to encompass the subsequent stages of plant harvesting and transportation. The plant factory's operational data will undergo annual collection and analysis.

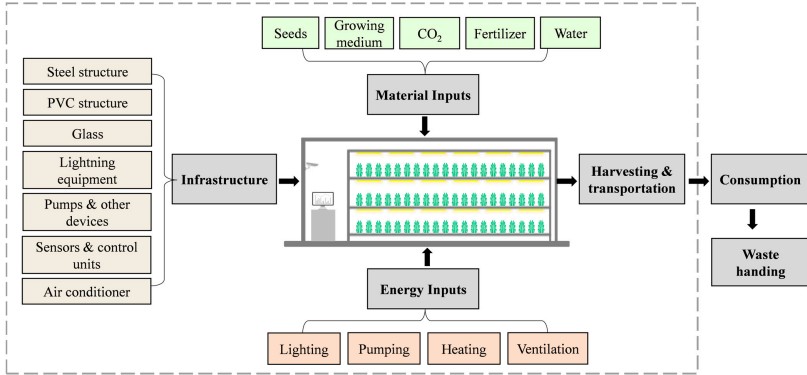

**Figure 11.** Illustration of the system boundary of plant factory LCA.

### 3.2.3. Functional Unit

The functional unit of a production system is a quantitative statement of the performance requirements that the system satisfies [65]. In previous studies of vertical farming, the standard functional unit frequently adopted for analysis is 1 kg of dry or fresh crops [36,48,66,67]. Following the previous research, the functional unit for LCA analysis in this study is defined as 1 kg dry *Pennisetum giganteum* production within a plant factory.

### 3.3. Life Cycle Inventory (LCI)

The inventory data for the plant factory are derived from experiments conducted under the optimal operational strategies detailed in Section 2.4. These strategies include 12 h of red–blue light exposure, temperature maintenance within the range of 10–35 °C, daily 4 min drip irrigation, fertilization every 5 days, and harvesting at intervals of 35–40 days. As for PGPF-EG (*Pennisetum giganteum* plant factory with exhaust gases introduced in an industrial park), it results in an annual yield of 2160 kg of fresh *Pennisetum giganteum*, equivalent to 496.8 kg of dry *Pennisetum giganteum*. The infrastructure data presented in this study are based on practical plant factory construction. The life cycle carbon emissions for the infrastructure are allocated over this life period. This study primarily focuses on the comprehensive assessment of materials, equipment, and transportation procedures within the framework of the plant factory. Ancillary aspects such as assembly and processing, which bear lesser relevance, have been intentionally omitted from the scope of analysis.

Table 9 presents the life cycle inventory of PGPF-EG. The functional unit is 1 kg of dry *Pennisetum giganteum*. The plant factory has the potential to generate 496.8 kg of dry *Pennisetum giganteum* annually. All metrics are proportionally allocated based on the yield achieved. In order to facilitate a more thorough investigation into the environmental efficacy of the plant factory, we have extended the analytical framework from PGPF-EG (*Pennisetum giganteum* plant factory with exhaust gases introduced) to include two supplementary *Pennisetum giganteum* growth scenarios, designated as PGPF (*Pennisetum giganteum* plant factory without exhaust gases introduced) and PGOF (*Pennisetum giganteum* growing in open fields). Apart from the abridged industrial exhaust treatment and introduction phases, the operational methodologies of PGPF and PGPF-EG are consistent. The *Pennisetum giganteum* cultivated in open fields is derived from previous experimental datasets, detailed inventory data for which can be found in Appendix A.

**Table 9.** Life cycle inventory of PGPF-EG. The functional unit is resource/kg of dry *Pennisetum giganteum*. The plant factory has the potential to generate 496.8 kg of dry *Pennisetum giganteum* annually. All metrics are proportionally allocated based on the yield achieved.

| Material | Quantity | Unit | Adapted Quantity | Functional Unit | Explanation |
|---|---|---|---|---|---|
| | | | Plant Factory Infrastructure | | |
| Steel pipe | 75 | kg | $3.77 \times 10^{-3}$ | kg/kg | Materials for plant racks (40-year life period) |
| Solar glass | 130.6 | kg | $6.57 \times 10^{-3}$ | kg/kg | Observation windows (40-year life period) |
| PVC calendered sheet | 244 | kg | $2.46 \times 10^{-2}$ | kg/kg | Materials for planting troughs and pipelines (20-year life period) |
| Light-emitting diode | 0.5 | kg | $1.01 \times 10^{-4}$ | kg/kg | Illumination equipment (10-year life period) |
| Machinery and computers | 3 | kg | $6.04 \times 10^{-4}$ | kg/kg | Pumps, valves, control cabinet, and other devices (10-year life period) |
| Transportation | 362.5 | tkm | $1.82 \times 10^{-2}$ | tkm/kg | Transportation of the plant factory to the industrial park |
| Land-use change | 18 | m$^2$ | $9.06 \times 10^{-4}$ | m$^2$/kg | Land-use change from industrial land to agricultural land for perennial crop |
| | | | *Pennisetum giganteum* Cultivating Operation (one-year cultivation period) | | |
| Soil | 403.2 | kg | 0.812 | kg/kg | Cultivation substrate in planting troughs |
| Stem segments | 10.2 | kg | $2.03 \times 10^{-2}$ | kg/kg | Stem segments of *Pennisetum giganteum* |
| CO$_2$ absorption from exhaust gases | 511.3 | kg | 1.03 | kg/kg | Including treatment process for industrial exhaust gases |
| CO$_2$ absorption from atmospheric environment | 235.2 | kg | 0.473 | kg/kg | Direct absorption from atmospheric environment |

**Table 9.** *Cont.*

| Material | Quantity | Unit | Adapted Quantity | Functional Unit | Explanation |
|---|---|---|---|---|---|
| Fertilizer | 1.1 | kg | $2.22 \times 10^{-3}$ | kg/kg | NPK (26-15-15) fertilizer |
| Reclaim water | 13.1 | m³ | $2.64 \times 10^{-2}$ | m³/kg | Including water treatment processes |
| Electricity consumption for lighting | 5026 | kWh | 10.1 | kWh /kg | Powered by photovoltaic solar energy from the industrial park. |
| Electricity consumption for other processes | 887 | kWh | 1.79 | kWh /kg | Including irrigation, sensors, and controller operation, introducing exhaust gases, etc. |
| Transportation | 185 | tkm | 0.372 | tkm/kg | Transportation of *Pennisetum giganteum* to processing sites |

### 3.4. Life Cycle Impact Assessment (LCIA)

This study focuses on a limited set of environmental indicators including GHG emissions, acidification and eutrophication impacts, abiotic resource depletion, and human toxicity. The aim is to conduct an environmental assessment using the life cycle impact assessment method ReCiPe 2016 [68]. Within the environmental indicators of ReCiPe, global warming (kg $CO_2$ eq), ozone formation (kg $NO_x$ eq), terrestrial acidification (kg $SO_2$ eq) and water consumption (m³) will be the primary focus in this study.

The stacked percentage bar chart of various environmental impact indicators is depicted in Figure 12, while the results for four crucial indicators are illustrated in Figure 13. As for the global warming indicator, among all sources of carbon emissions, the lighting phase accounts for a significant 67% of total emissions, followed by other processes (including air conditioning, irrigation, fan operation, etc.) and the infrastructure, accounting for 15% and 10%, respectively. The lighting phase also plays a central role in the ozone formation indicator and the terrestrial acidification indicator, contributing to 66% and 62%, respectively. As for the water consumption indicator, the main contributors are reclaimed water and the lighting phase, contributing 44% and 43%, respectively. The detailed results are provided in Appendix B.

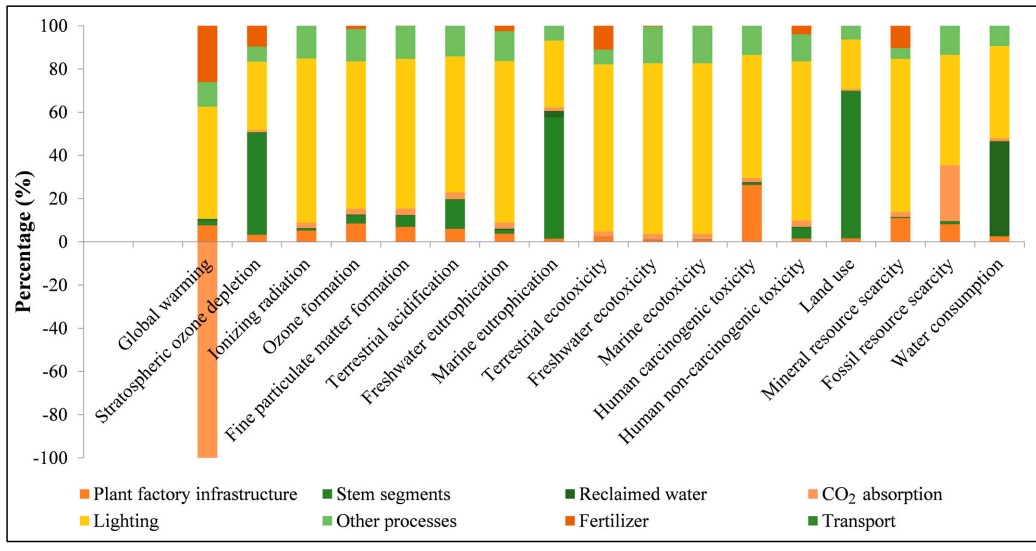

**Figure 12.** Life cycle environmental indicator contribution percentages of PGPF-EG.

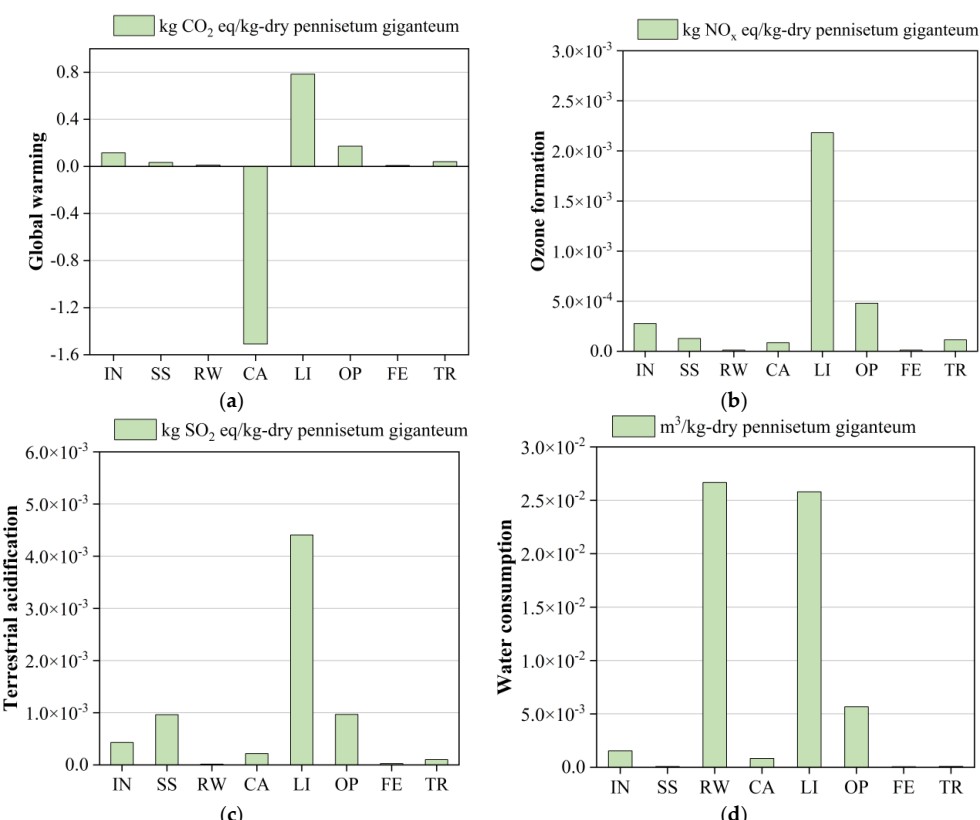

**Figure 13.** Various life cycle environmental characterization indicators of PGPF-EG: (**a**) global warming; (**b**) ozone formation; (**c**) terrestrial acidification; (**d**) water consumption. (IN: infrastructure; SS: stem segments; RW: reclaimed water; CA: $CO_2$ absorption; LI: lighting; OP: other processes; FE: fertilizer; TR: transport.)

The environmental benefits compared between PGPF-EG and PGPF are illustrated in Figure 14. In PGPF, the production of 1 kg of dry *Pennisetum giganteum* sequestered 0.27 kg of net $CO_2$ equivalent, whereas in PGPF-EG this value increased to 0.35 kg of net $CO_2$ equivalent, representing a 29% increase. Accompanied by a 15% improvement in the yield of *Pennisetum giganteum* for PGPF-EG, the overall $CO_2$ capture capacity of the plant factory was correspondingly increased by a substantial 50%.

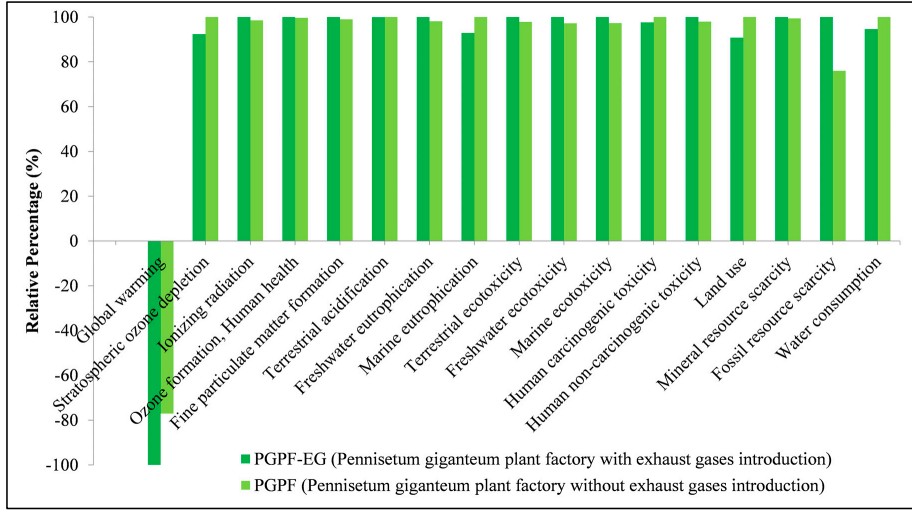

**Figure 14.** Life cycle environmental characterization indicators comparison of 1 kg dry *Pennisetum giganteum* production under different environmental conditions.

The environmental impact of the plant factory under different electricity generation sources is illustrated in Figures 15 and 16. As expected, compared to coal and natural gas, clean energy sources exhibit significantly fewer environmental impacts for most environmental assessment metrics such as global warming, ozone formation, fine particulate matter formation, and freshwater eutrophication. Plant factories powered by coal and natural gas fail to achieve carbon sequestration and, on the contrary, exhibit significant carbon emissions. This indicates that for carbon sequestration through a plant factory, the electricity must be sourced from low-carbon energy sources like photovoltaics, wind, nuclear power, and hydropower.

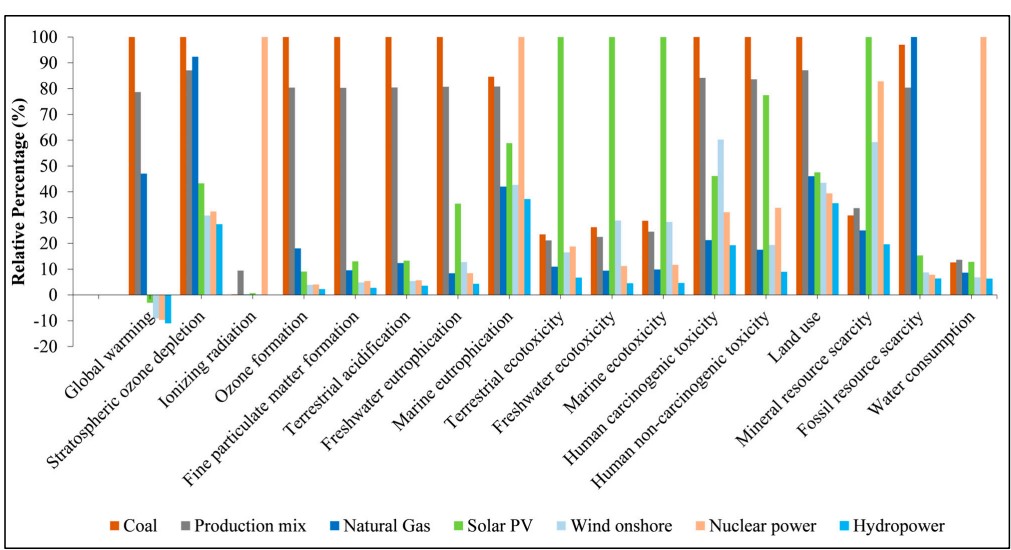

**Figure 15.** Life cycle environmental characterization indicators of 1 kg dry *Pennisetum giganteum* production from a plant factory using different electricity generation sources.

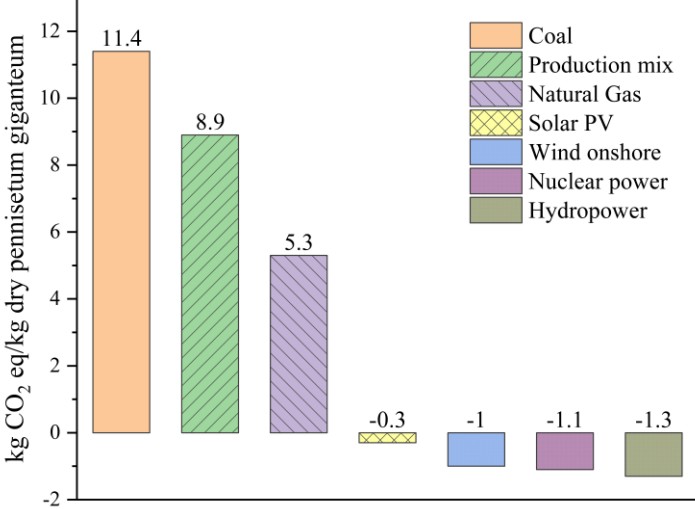

**Figure 16.** Comparative analysis of $CO_2$ eq emission associated with 1 kg of dry *Pennisetum giganteum* cultivated in a plant factory using different electricity generation sources.

### 3.5. Interpretation of Results

The life cycle assessment results presented above suggested that carbon emissions from the plant factory predominantly stem from energy consumption during the growth phases of these plants, among which the lighting contributes to the most substantial electricity consumption. Hence, the strategic selection of energy-efficient illumination equipment, combined with the implementation of scientifically optimized lighting strategies, holds

significant promise in reducing carbon emissions throughout the operational duration of the plant factory. Likewise, the adoption of materials with reduced carbon emissions and the implementation of more efficient design schemes can effectively mitigate infrastructure-related carbon emissions. In addition, plant factories require renewable electricity to fulfill their role as carbon sinks. Meanwhile, optimizing the lighting strategy is anticipated to have a positive impact on both the ozone formation indicator and the terrestrial acidification indicator. As for water consumption, the implementation of more precise irrigation strategies, along with the optimization of water treatment processes within the industrial park, is expected to reduce the water consumption indicator of the plant factory.

In comparison to PGPF, PGPF-EG exhibits a 29% increase in the net $CO_2$ sequestered per 1 kg of dry *Pennisetum giganteum*. Simultaneously, PGPF-EG increases the *Pennisetum giganteum* yield by 15%, resulting in a 50% overall increase in the annual $CO_2$ sequestration capacity of the plant factory. Based on the results of the LCA, a plant factory in an industrial park (with dimensions of 6 m × 3 m × 2.8 m) has the potential to annually produce a maximum of 2160 kg of *Pennisetum giganteum* while sequestering 746.5 kg of $CO_2$. After offsetting the carbon emissions associated with the plant factory's infrastructure and energy consumption, there is an annual reduction of 174 kg in $CO_2$ emissions. These findings collectively underscore the promising prospects of *Pennisetum giganteum* plant factories as a sustainable and eco-efficient means of biomass production within industrial parks.

Furthermore, the annual carbon sequestration performance of *Pennisetum giganteum* within a 1 m$^2$ area under different conditions is illustrated in Figure 17 for comparative analysis. These values are derived from a combination of yield data and results from the LCA. In a 1 m$^2$ area, *Pennisetum giganteum* sequesters 6.2 kg of $CO_2$ annually in an open-field environment, while for a plant factory with industrial exhaust gas introduction, it sequesters 9.7 kg of $CO_2$, resulting in a 56% increase. Moreover, the plant factory system is well suited for scalable operations. Taking into account structural load-bearing capacity and safety standards, it is feasible to increase the plant factory's height by up to five times its current dimensions, which significantly expands the available planting area. Under these assumptions, the plant factory can achieve an annual carbon sequestration rate of 52.1 kg/m$^2$, equivalent to 521 tons of carbon sequestration per hectare per year, which is 8.4 times greater than that of open-field cultivation.

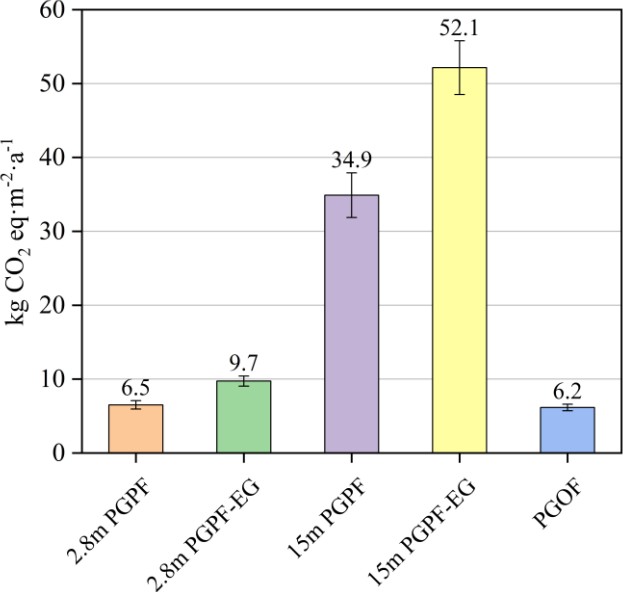

**Figure 17.** A comparative analysis of annual carbon sequestration per unit area under various conditions (2.8 m and 15 m represent the height of the plant factory; the error bars represent the results of data collected from five experimental groups).

## 4. Conclusions

This study thoroughly investigated the feasibility of carbon capture by an intelligent plant factory within an industrial park. A life cycle assessment was conducted to quantify the environmental impact and carbon benefits under various scenarios.

An automated and intelligent vertical plant factory was designed and established in an industrial park, utilizing high-$CO_2$ concentration exhaust gases from an automotive painting workshop as the carbon source. Renewable electricity and reclaimed water were integrated into the plant factory system. Noteworthy features include autonomous or remote control over lighting, air conditioning, irrigation, data collection, etc., making it suitable for unmanned management. *Pennisetum giganteum* was chosen as the carbon capture medium for its exceptional carbon sequestration capacity, rapid growth rate, and manageable cultivation requirements.

The research identified an optimal cultivation strategy for *Pennisetum giganteum* through a series of planting experiments. This strategy involves implementing a growth cycle ranging from 35 to 40 days, enabling up to 10 harvests annually. Key factors for achieving high *Pennisetum giganteum* yields include maintaining a daily 12 h of red–blue light exposure, controlling temperatures within the range of 10–35 °C, and implementing a daily 4 min drip irrigation. The introduction of industrial exhaust gases, maintaining $CO_2$ concentrations at 1600–2000 ppm, resulted in a 15% increase in *Pennisetum giganteum* yields, showcasing the plant factory's ability to capture $CO_2$ from exhaust gases to enhance plant growth.

The carbon capture capacity and environmental impact have been quantified through life cycle assessment. Our analysis reveals that the primary carbon emission arises from the lighting phase, accounting for 67% of the total emissions. Other processes (including air conditioning, irrigation, fan operation, etc.) and infrastructure contribute 15% and 10%, respectively. Under the aforementioned cultivation strategies, supplied with industrial exhaust gases and photovoltaic power, the plant factory with the dimensions 3 m × 6 m × 2.8 m can annually produce 2160 kg of fresh *Pennisetum giganteum*, reducing carbon emissions by 174 kg, with the annual carbon sequestration per unit area increased by 56% compared to open-field cultivation. Expanding the plant factory's height up to 15 m can lead to an annual carbon sequestration rate of 52.1 kg/m$^2$, equivalent to 521 tons of carbon sequestration per hectare per year. This is 8.4 times greater than that of open-field cultivation, suggesting that large-scale plant factories have the potential to neutralize the carbon emissions of an entire industrial park.

**Author Contributions:** Conceptualization, X.D. and J.L.; data curation, H.C., N.Z. and Q.W.; investigation, H.C.; methodology, H.C.; project administration, J.Y.; resources, J.Y.; software, N.Z.; supervision, X.D.; validation, J.L. and Z.S.; writing—original draft, H.C. and X.D.; writing—review and editing, X.D., J.L., N.Z., Q.W. and Z.S. All authors have read and agreed to the published version of the manuscript.

**Funding:** This research was financially supported by the Volvo industry–university collaboration fund, and it was also partially supported by the National Natural Science Foundation of China under Grant No. 52376160, the National Natural Science Foundation of China under Grant No. 52006137, and the Shanghai Sailing Program under Grant No. 19YF1423400.

**Institutional Review Board Statement:** Not applicable.

**Informed Consent Statement:** Not applicable.

**Data Availability Statement:** The data presented in this study are available on request from the corresponding author.

**Conflicts of Interest:** Authors Jie Lei, Ning Zhang, Qianrui Wang, Zhiang Shi and Jinxing Yang were employed by the company Volvo Cars Technology (Shanghai) Co., Ltd. The remaining authors declare that the research was conducted in the absence of any commercial or financial relationships that could be construed as a potential conflict of interest.

## Appendix A

This appendix provides LCI for PGOF (*Pennisetum giganteum* in open fields) in Section 3.3. Based on the experimental results, *Pennisetum giganteum* can be harvested twice a year, with an annual yield of 31.3 kg/m$^2$, resulting in 7.2 kg of dry *Pennisetum giganteum*.

**Table A1.** Life cycle inventory of dry *Pennisetum giganteum* production for PGOF.

| Material | Quantity | Unit | Adapted Quantity | Functional Unit | Explanation |
| --- | --- | --- | --- | --- | --- |
| Stem segments | 0.22 | kg | $3.06 \times 10^{-2}$ | kg/kg | Stem segments of *Pennisetum giganteum* |
| $CO_2$ absorption | 10.8 | kg | 1.50 | kg/kg | Direct absorption from atmospheric environment |
| Fertilizer | 0.02 | kg | $2.78 \times 10^{-3}$ | kg/kg | NPK (26-15-15) fertilizer |
| Irrigation | 0.51 | kg | $7.08 \times 10^{-2}$ | kg/kg | The water used for irrigating the plants |
| Electricity consumption | 2.78 | kWh | $3.86 \times 10^{-1}$ | kWh/kg | Powered by production mix electricity for operation including planting, pumping, and harvesting |
| Land-use change | 1 | m$^2$ | $1.39 \times 10^{-1}$ | m$^2$/kg | Land-use change from industrial land to agricultural land for perennial crop |
| Transportation | 2.68 | tkm | $3.72 \times 10^{-1}$ | tkm/kg | Transportation of *Pennisetum giganteum* to processing sites |

## Appendix B

This appendix provides LCA results for PGPF-EG (*Pennisetum giganteum* plant factory with exhaust gases introduced in an industrial park) in Section 3.4 with the method ReCiPe 2016 Midpoint (H).

**Table A2.** Indicator results of LCA for the scenario of PGPF-EG.

| Impact Category | Unit | Infrastructure | Stem Segments | Reclaimed Water | CO$_2$ Absorption | Lighting | Other Processes | Fertilizer | Transport, Lorry | Total |
|---|---|---|---|---|---|---|---|---|---|---|
| Global warming | kg CO$_2$ eq | $1.1 \times 10^{-1}$ | $3.4 \times 10^{-2}$ | $1.1 \times 10^{-2}$ | $-1.5$ | $7.8 \times 10^{-1}$ | $1.7 \times 10^{-1}$ | $7.9 \times 10^{-3}$ | $4.0 \times 10^{-2}$ | $-3.5 \times 10^{-1}$ |
| Stratospheric ozone depletion | kg CFC11 eq | $3.7 \times 10^{-8}$ | $5.4 \times 10^{-7}$ | $9.7 \times 10^{-10}$ | $1.2 \times 10^{-8}$ | $3.6 \times 10^{-7}$ | $7.8 \times 10^{-8}$ | $9.4 \times 10^{-8}$ | $2.6 \times 10^{-8}$ | $1.1 \times 10^{-6}$ |
| Ionizing radiation | kBq Co-60 eq | $4.0 \times 10^{-3}$ | $8.5 \times 10^{-4}$ | $6.8 \times 10^{-5}$ | $1.8 \times 10^{-3}$ | $5.6 \times 10^{-2}$ | $1.2 \times 10^{-2}$ | $6.6 \times 10^{-5}$ | $7.1 \times 10^{-4}$ | $7.6 \times 10^{-2}$ |
| Ozone formation, human health | kg NOx eq | $2.8 \times 10^{-4}$ | $1.3 \times 10^{-4}$ | $1.2 \times 10^{-5}$ | $8.5 \times 10^{-5}$ | $2.2 \times 10^{-3}$ | $4.8 \times 10^{-4}$ | $1.2 \times 10^{-5}$ | $1.1 \times 10^{-4}$ | $3.3 \times 10^{-3}$ |
| Fine particulate matter formation | kg PM2.5 eq | $2.0 \times 10^{-4}$ | $1.5 \times 10^{-4}$ | $6.3 \times 10^{-6}$ | $8.6 \times 10^{-5}$ | $2.0 \times 10^{-3}$ | $4.4 \times 10^{-4}$ | $8.8 \times 10^{-6}$ | $4.4 \times 10^{-5}$ | $2.9 \times 10^{-3}$ |
| Terrestrial acidification | kg SO$_2$ eq | $2.8 \times 10^{-4}$ | $1.3 \times 10^{-4}$ | $1.2 \times 10^{-5}$ | $8.9 \times 10^{-5}$ | $2.3 \times 10^{-3}$ | $5.0 \times 10^{-4}$ | $1.2 \times 10^{-5}$ | $1.2 \times 10^{-4}$ | $3.4 \times 10^{-3}$ |
| Freshwater eutrophication | kg P eq | $4.3 \times 10^{-4}$ | $9.6 \times 10^{-4}$ | $1.4 \times 10^{-5}$ | $2.2 \times 10^{-4}$ | $4.4 \times 10^{-3}$ | $9.7 \times 10^{-4}$ | $2.5 \times 10^{-5}$ | $1.0 \times 10^{-4}$ | $7.1 \times 10^{-3}$ |
| Marine eutrophication | kg N eq | $3.3 \times 10^{-5}$ | $1.4 \times 10^{-5}$ | $5.5 \times 10^{-6}$ | $2.4 \times 10^{-5}$ | $6.3 \times 10^{-4}$ | $1.4 \times 10^{-4}$ | $1.6 \times 10^{-6}$ | $9.2 \times 10^{-6}$ | $8.6 \times 10^{-4}$ |
| Terrestrial ecotoxicity | kg 1,4-DCB | $2.5 \times 10^{-6}$ | $1.0 \times 10^{-4}$ | $5.4 \times 10^{-6}$ | $2.9 \times 10^{-6}$ | $5.6 \times 10^{-5}$ | $1.2 \times 10^{-5}$ | $1.8 \times 10^{-7}$ | $3.6 \times 10^{-7}$ | $1.8 \times 10^{-4}$ |
| Freshwater ecotoxicity | kg 1,4-DCB | $8.0 \times 10^{-1}$ | $7.0 \times 10^{-2}$ | $6.0 \times 10^{-3}$ | $9.2 \times 10^{-1}$ | $29$ | $6.3$ | $2.2 \times 10^{-2}$ | $4.5 \times 10^{-1}$ | $37$ |
| Marine ecotoxicity | kg 1,4-DCB | $3.7 \times 10^{-3}$ | $9.5 \times 10^{-4}$ | $1.0 \times 10^{-4}$ | $1.3 \times 10^{-2}$ | $3.8 \times 10^{-1}$ | $8.4 \times 10^{-2}$ | $2.2 \times 10^{-4}$ | $1.2 \times 10^{-3}$ | $4.9 \times 10^{-1}$ |
| Human carcinogenic toxicity | kg 1,4-DCB | $5.1 \times 10^{-3}$ | $1.2 \times 10^{-3}$ | $1.4 \times 10^{-4}$ | $1.6 \times 10^{-2}$ | $4.9 \times 10^{-1}$ | $1.1 \times 10^{-1}$ | $2.9 \times 10^{-4}$ | $1.9 \times 10^{-3}$ | $6.2 \times 10^{-1}$ |
| Human non-carcinogenic toxicity | kg 1,4-DCB | $5.3 \times 10^{-2}$ | $1.5 \times 10^{-3}$ | $1.1 \times 10^{-3}$ | $3.7 \times 10^{-3}$ | $1.1 \times 10^{-1}$ | $2.5 \times 10^{-2}$ | $2.5 \times 10^{-4}$ | $2.5 \times 10^{-3}$ | $2.0 \times 10^{-1}$ |
| Land use | m$^2$a crop eq | $7.2 \times 10^{-2}$ | $2.4 \times 10^{-1}$ | $1.0 \times 10^{-2}$ | $1.4 \times 10^{-1}$ | $3.4$ | $7.6 \times 10^{-1}$ | $4.6 \times 10^{-3}$ | $3.7 \times 10^{-2}$ | $4.7$ |
| Mineral resource scarcity | kg Cu eq | $1.7 \times 10^{-3}$ | $7.4 \times 10^{-2}$ | $9.7 \times 10^{-5}$ | $7.9 \times 10^{-4}$ | $2.5 \times 10^{-2}$ | $5.5 \times 10^{-3}$ | $1.2 \times 10^{-4}$ | $1.5 \times 10^{-3}$ | $1.1 \times 10^{-1}$ |
| Fossil resource scarcity | kg oil eq | $2.1 \times 10^{-3}$ | $8.7 \times 10^{-5}$ | $5.9 \times 10^{-6}$ | $4.3 \times 10^{-4}$ | $1.3 \times 10^{-2}$ | $3.0 \times 10^{-3}$ | $4.5 \times 10^{-5}$ | $1.2 \times 10^{-4}$ | $1.9 \times 10^{-2}$ |
| Water consumption | m$^3$ | $3.3 \times 10^{-2}$ | $5.0 \times 10^{-3}$ | $8.9 \times 10^{-4}$ | $1.0 \times 10^{-1}$ | $2.0 \times 10^{-1}$ | $4.5 \times 10^{-2}$ | $1.1 \times 10^{-3}$ | $1.3 \times 10^{-2}$ | $4.1 \times 10^{-1}$ |

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
