# Peer review of "Life Cycle Assessment of Carbon Capture by an Intelligent Vertical Plant Factory within an Industrial Park"

_sustainability, doi:10.3390/su16020697_

Round 1

Reviewer 1 Report

Comments and Suggestions for Authors

The paper analyzes experimental data on the use of plants to capture carbon. It has interesting results and discussions to be published. Below are some suggestions that I think could improve the paper analysis's quality.

- Include more detailed information about the instrumentation used in the measurements and the procedures used to obtain the results.

- For the results, only the averages are presented. I suggest including information that shows the dispersion of the results, such as the standard deviation (in the tables and graphs where the results are represented).

- I suggest applying statistical tests to prove the statement that there are significant differences between the results obtained.

Reviewer 2 Report

Comments and Suggestions for Authors

The authors list numerous intriguing applications of this plant. Of particular interest are methane conversion and hydrogen production. Please provide a more detailed description of the plant's role in methane conversion and hydrogen production

Reviewer 3 Report

Comments and Suggestions for Authors

This study introduces an attractive approach to reducing carbon emissions. The paper is well written and the logic is strong. I only have one concern about the advantages and disadvantages of the proposed method compared with other carbon-removal technologies, such as geological sequestration, chemical conversion, etc. It is suggested to add some discussion or/and a table to highlight the comparison.

Reviewer 4 Report

Comments and Suggestions for Authors

Dear authors of the manuscript Life Cycle Assessment of Carbon Capture by a Smart Vertical Plant Factory, I want to mention that your work can be accepted after some minor revisions:
1. Images and graphics must improve their resolution
2. The conclusion must be rewritten to match the objectives of the study and should not be in list format.

Round 2

Reviewer 1 Report

Comments and Suggestions for Authors

The authors solved most of the issues raised.

With the standard deviation, some conclusions, such as one result being larger than the other, do not seem to have statistical power. In other words, although the average is larger, due to dispersion, it cannot be said that there are significant differences. I believe that simple hypothesis tests, such as the t-test or ANOVA, would enhance the analysis of the results. Using Table 5 as an example, there do not appear to be statistically significant differences between the heights of groups a, c, and d.

There are incorrect units used (k for kilo is lower case, W for Watts is upper case)

All figures and tables must be cited and described in the text. Please check because there are some cases along the paper.
